# Next-Generation Genome-Scale Metabolic Modeling through Integration of Regulatory Mechanisms

**DOI:** 10.3390/metabo11090606

**Published:** 2021-09-07

**Authors:** Carolina H. Chung, Da-Wei Lin, Alec Eames, Sriram Chandrasekaran

**Affiliations:** 1Department of Biomedical Engineering, University of Michigan, Ann Arbor, MI 48109, USA; chechung@umich.edu (C.H.C.); eamesa@umich.edu (A.E.); 2Department of Computational Medicine and Bioinformatics, University of Michigan, Ann Arbor, MI 48109, USA; daweilin@umich.edu; 3Program in Chemical Biology, University of Michigan, Ann Arbor, MI 48109, USA; 4Center for Bioinformatics and Computational Medicine, University of Michigan, Ann Arbor, MI 48109, USA; 5Rogel Cancer Center, University of Michigan Medical School, Ann Arbor, MI 48109, USA

**Keywords:** metabolic regulation, metabolic networks, constraint-based modeling, systems biology, genome-scale network models

## Abstract

Genome-scale metabolic models (GEMs) are powerful tools for understanding metabolism from a systems-level perspective. However, GEMs in their most basic form fail to account for cellular regulation. A diverse set of mechanisms regulate cellular metabolism, enabling organisms to respond to a wide range of conditions. This limitation of GEMs has prompted the development of new methods to integrate regulatory mechanisms, thereby enhancing the predictive capabilities and broadening the scope of GEMs. Here, we cover integrative models encompassing six types of regulatory mechanisms: transcriptional regulatory networks (TRNs), post-translational modifications (PTMs), epigenetics, protein–protein interactions and protein stability (PPIs/PS), allostery, and signaling networks. We discuss 22 integrative GEM modeling methods and how these have been used to simulate metabolic regulation during normal and pathological conditions. While these advances have been remarkable, there remains a need for comprehensive and widespread integration of regulatory constraints into GEMs. We conclude by discussing challenges in constructing GEMs with regulation and highlight areas that need to be addressed for the successful modeling of metabolic regulation. Next-generation integrative GEMs that incorporate multiple regulatory mechanisms and their crosstalk will be invaluable for discovering cell-type and disease-specific metabolic control mechanisms.

## 1. Introduction

Cellular metabolism is a fundamental biological process used by all living organisms to generate and expend energy for growth [1]. Although metabolism functions the same way between different organisms, its regulation is dictated by perturbations and environments encountered by each individual organism [2]. Metabolic regulation entails the coordination between distinct yet interconnected mechanisms that control enzymatic activity and abundance, thereby modulating fluxes through metabolic reactions [3]. For instance, transcriptional regulation dictates enzyme abundance via changes in gene expression in response to nutrient availability. This form of regulation usually operates over a long timescale (e.g., hours) and defines a general range for fluxes [4]. Then, post-translational regulation may occur in succession, where enzymatic levels and activity are fine-tuned over a shorter timescale (e.g., milliseconds) [5]. Post-translational regulation may involve chemical modifications, ligand binding, or interaction with other proteins that influence other downstream events (e.g., cell signaling).

Most of our understanding of metabolism comes from cumulative data from biochemical experiments carried out over the past century. Although experimental evidence has been insightful, this type of investigation is often restricted to studying an isolated metabolic pathway in a limited number of conditions. To overcome this limitation, modern approaches seek to study metabolism from a systems biology perspective [6,7,8]. In recent decades, genome-scale metabolic models (GEMs) have emerged as a powerful tool for this purpose, enabling the elucidation of complex systems-level metabolism. GEMs are computational representations of metabolic networks accounting for the entirety of metabolic activity encoded in the genome for a given organism [9]. Principally, they involve a set of mass-balanced metabolic reactions and metabolites represented in a stoichiometric matrix. GEMs also include gene–protein reaction (GPR) associations describing the relationship between thousands of genes, proteins, and reactions [10]. GEMs facilitate quantitative, in silico simulations of how environmental and genetic changes influence cellular metabolism [11]. Since their introduction, over 6000 GEMs have been reconstructed across bacteria, archaea, and eukarya [12].

While GEMs in their most basic form can model cellular metabolism reasonably well [13,14,15], they are limited by the inability to account for the regulatory mechanisms fundamental to cellular metabolism [15,16,17,18,19]. Since the scope and capabilities of GEMs are a function of how accurately they emulate metabolism, a frontier of metabolic modeling is developing ways to integrate regulatory aspects into GEMs. In this review, we cover six different types: transcriptional regulatory networks (TRNs), post-translational modifications (PTMs), epigenetics, protein–protein interactions and protein stability (PPIs/PS), allostery, and signaling (Figure 1).

While this enumeration of regulatory mechanisms is not exhaustive, these six mechanisms represent major ways through which cellular regulation affects metabolism. Of note, various methods have focused on integrating omics data directly into GEMs to improve model accuracy. Several excellent reviews have already covered this topic in greater detail [20,21,22,23]; thus, such methodologies will not be the focus in this review. Instead, we present and discuss 22 methods that have mechanistically integrated regulatory information relevant to the six mechanisms listed above into GEMs (Table 1).

## 2. Modeling of Metabolic Regulation

### 2.1. Simulating Metabolic Networks Using Constraint-Based Modeling (CBM)

Before discussing how regulatory mechanisms have been integrated into GEMs, we begin with a brief overview on mathematical methods relevant to the algorithms discussed throughout this review (Figure 2).

Constraint-based modeling (CBM) is the standard framework for reconstructing and analyzing GEMs, primarily through the addition of model constraints [11]. The most basic form of CBM is flux balance analysis (FBA) [46]. FBA begins with the stoichiometric matrix *S*, where rows signify metabolites and columns represent reactions. For each reaction (column), the stoichiometric coefficients of all metabolites involved in the reaction are set to a non-zero value. Specifically, positive coefficients indicate the production of a metabolite, while negative coefficients indicate the consumption of a metabolite. A key assumption of FBA is steady-state metabolism in which each metabolite’s production and consumption is balanced equally [47]. Mathematically, FBA aims to simulate reaction fluxes at steady state, which leads to solving the following system of equations (Equation (1)):(1)S×v=0
where *S* represents the stoichiometric matrix, *v* is the vector of reaction fluxes, and *b*, representative of changes in metabolite concentrations, is set to a zero vector to mathematically reflect steady-state metabolism. Since the number of reactions exceeds the number of metabolites, the system is underdetermined, and a large solution space exists; however, this can be narrowed by imposing additional constraints and narrowing the flux bounds. In FBA, linear optimization techniques are applied to solve for a flux distribution that optimizes an objective function (*Z_obj_*). To reflect evolutionary pressure, growth is typically maximized by defining a biomass objective function that consists of biomass precursors [48]. Other objectives such as the maximization of ATP can also be used in FBA [49]. These constraints are mathematically represented as follows in Equation (2a) and Equation (2b):(2a)$$Maximize\;Z_{obj}=v_{biomass}$$
(2b)lb≤v≤ub
where *Z_obj_* is the objective function, *v_biomass_* is the flux through a user-defined biomass reaction, *v* is the vector of all reaction fluxes, *lb* is a vector of the lower bound flux limit, and *ub* is a vector of the upper bound flux limit. 

Although FBA can generate flux solutions that match experimental data [13,49], this method may still generate non-unique solutions. This predicament arises as multiple combinations of reaction fluxes can satisfy the constraints and lead to the same objective. To account for this uncertainty, flux variability analysis (FVA) aims to capture the entire feasible space of flux solutions. FVA introduced two additional linear programming (LP) problems (mathematically defined below) [50] to be imposed along with the FBA constraints listed above:(3a)$$Minimize\;v_{i}\;where\;i=1,\;\ldots ,\;n$$
$$s.t.\;Eqs.\;1,\;2.1,\;2.2\;and\;lb_{i}\leq v_{i}\leq ub_{i}$$
(3b)$$Maximize\;v_{i}\;where\;i=1,\;\ldots ,\;n$$
$$s.t.\;Eqs.\;1,\;2.1,\;2.2\;and\;lb_{i}\leq v_{i}\leq ub_{i}$$
where *v_i_* represents the flux through reaction *i*, *n* is equivalent to the total number of reactions, *lb_i_* is the lower flux bound for reaction *i*, and *ub_i_* is the upper flux bound for reaction *i*. In FVA, the objective function (*Z_obj_*) serves as a reference to find the possible lower and upper flux limits for Equation (3a) and Equation (3b), respectively. A different method, called parsimonious FBA (pFBA), also improves GEM simulation accuracy by determining the feasible flux ranges based on the assumption that cells maximize efficient enzyme usage to promote their growth [51]. This assumption is mathematically imposed by minimizing the overall flux in the system (Equation (4)):(4)$$Minimize\;\overset{n}{\underset{i}{\sum }}v_{i}s.t.\;v_{biomass}=v_{biomass,\;lb}$$
$$s.t.\;Eqs.\;1,\;2.1,\;2.2\;and\;lb_{i}\leq v_{i}\leq ub_{i}$$
where *v_i_* is the flux through reaction *i*, *n* is the total number of reactions, *v_biomass_* is the flux through the biomass reaction, *v_biomass,lb_* is the lower limit of the flux through the biomass reaction, *lb_i_* is the lower flux bound for reaction *i*, and *ub_i_* is the upper flux bound for reaction *i*. Of note, the flux solutions achieved using pFBA tend to emphasize a small number of high-flux reactions due to the underlying assumptions of pFBA.

### 2.2. Transcriptional Regulatory Networks (TRNs)

Transcriptional regulation describes the diverse ways in which gene transcription into messenger RNA (mRNA) is regulated by the cell. mRNA transcripts carry gene-encoded instructions for protein production to the ribosome, where translation occurs. Since gene transcription is required to express the proteins that participate in metabolic reactions, the association between genes, proteins, and reactions is strongly dependent on transcriptional regulation. Transcriptional regulation is often represented through transcriptional regulatory networks (TRNs), which describe how the transcription of target genes is influenced by regulatory genes and environmental perturbations [24,52]. In the past several decades, knowledge of transcriptional regulation has expanded substantially, and transcriptional dysregulation has been found to be implicated in a wide range of diseases [53,54]. Methods integrating TRNs into GEMs can be broadly categorized based on the granularity in transcription factor (TF)–target gene relationships. Following this trend, the following discussion is partitioned between the integration of discrete versus continuous TRNs.

#### 2.2.1. Boolean TRNs

The simplest TRN is a Boolean network in which genes assume one of two states: active or inactive. This status is based on multiple factors including the activity of regulating genes and environmental conditions [52]. Boolean TRNs were first integrated with GEMs in *regulatory flux balance analysis* (rFBA). This algorithm is an expansion of FBA in which metabolic models with Boolean TRNs are used to predict steady-state flux distributions over a series of time intervals [24]. To model transcriptional regulation, the activity of genes in a given time interval is regulated by the metabolic state of the previous interval. Using rFBA, the authors improved the accuracy in *E. coli* growth predictions from GEMs across numerous metabolic environments.

While rFBA marked an important step forward in modeling the transcriptional regulation of metabolism, it is limited by its selection of only one valid metabolic state for each interval when multiple could occur. This limitation was addressed by *steady-state rFBA* (SR-FBA), which simulates steady-state metabolism more comprehensively by incorporating both metabolic and regulatory constraints into a single optimization problem [25]. Instead of running a series of flux solutions (as done in rFBA), SR-FBA calculates fluxes for a single time step using mixed-integer linear programming (MILP). It also quantitatively determines the influence of regulatory networks versus metabolic constraints in simulating metabolism by comparing SR-FBA fluxes to those generated by standard FBA. While SR-FBA succeeds at precluding inconsistent regulatory constraints, it fails to account for metabolic transitions and feedback loops as a direct result. Another limitation of both rFBA and SR-FBA is a failure to predict internal metabolite concentrations. This was partially addressed by *integrated FBA* (iFBA), which incorporates kinetic models and ordinary differential equations (ODEs) [27]; however, sparseness in kinetic parameter availability renders iFBA infeasible to implement at the genome scale.

In order to facilitate the combination of Boolean TRNs and GEMs, a platform called *toolbox for integrating genome-scale metabolism, expression, and regulation* (TIGER) was developed [29]. TIGER automates the combination of GEMs, TRNs, and transcriptomics into a single regulatory-metabolic model. A key feature of TIGER is that it converts a list of Boolean rules and GPRs into a set of linear inequalities. *FlexFlux* is a similar platform but improves upon TIGER in terms of usability as TRNs and GEMs can be inputted in a standardized file format [30]. *FlexFlux* also enables the integration of multi-state regulatory networks and allows the conversion of discrete regulatory states into continuous intervals. 

#### 2.2.2. Continuous TRNs

While Boolean TRNs have been shown to improve flux simulations, there are several limitations endemic to this framework. These include an overly simplistic model of metabolism, a theory-driven approach, and the need for extensive literature searching to define regulatory rules [24]. This was partially addressed by Lee et al. (2007), wherein rFBA was expanded from binarily describing regulation to assigning one of eight discrete weights to regulatory interactions, yielding more accurate predictions for certain gene expression levels [26]. *Probabilistic regulation of metabolism* (PROM) is a method that more comprehensively addressed these limitations by shifting the discrete paradigm to a continuous model of transcriptional regulation [28]. Using TF–target relationships and transcriptomics data, PROM builds a continuous TRN where target gene expression is controlled via continuous probabilities based on the expression of regulatory genes, and fluxes are continuously restricted (Figure 3). A further advantage of PROM is its automated integration of gene expression data for building TRNs. PROM 2.0 applied the original method to construct a continuous TRN for *Mycobacterium tuberculosis* (*M. tb*), leading to enhanced simulation accuracy using the *M. tb* GEM [31].

Advancing the continuous TRN paradigm, several other algorithms have been developed. *Transcriptional regulated FBA* (TRFBA) is a method that represents TRNs continuously and improves upon PROM by reducing the amount of transcriptomics data required to build continuous TRNs [34]. Of note, TRFBA introduces two continuous constraints: one that restricts gene-associated reactions based on expression levels of that gene and one that correlates target gene expression levels with regulatory gene expression levels. TRFBA has shown improvements to PROM, particularly for growth rate predictions. Benchmarking PROM and TRFBA against pFBA reveals that while both TRN methods exceed simulation accuracy for *E. coli*, only TRFBA is more accurate than pFBA when applied to *S. cerevisiae* [34]. However, a major limitation of both PROM and TRFBA is the requirement of pre-defined TF–target gene relationships. *CoRegFlux* was developed to address this limitation [32]. Using *learning cooperative regulation networks* (hLICORN), *CoRegFlux* determines TF–target gene relationships from transcriptomics data in the absence of a pre-existing TRN before predicting fluxes.

*Integrated deduced and metabolism* (IDREAM) is a method that functions similarly to continuous TRN methods such as PROM but alternatively integrates an *environment and gene regulatory influence network* (EGRIN) [33,55]. EGRIN provides information on environmental- and condition-specific effects on gene regulation to better capture the regulatory influence of TFs on their target genes. IDREAM has shown improvements to PROM, especially when simulating metabolism for eukaryotic cells. Across multiple yeast models and environmental conditions, IDREAM is more accurate than PROM in predicting growth rates [33]. Building off of IDREAM and regulatory-metabolic models, *optimization of regulatory and metabolic networks* (OptRAM) is an in silico strain design algorithm that optimizes the production of desired metabolites and has successfully been applied to *S. cerevisiae* [35].

While TRN modeling and integration into GEMs has advanced considerably, these integration methods fail to account for additional regulatory mechanisms such as signaling or allostery. This limits the extent to which they accurately model cellular metabolism, as significant numbers of metabolic fluxes are unable to be explained by transcriptional regulation alone [56], necessitating the integration of additional forms of regulation into GEMs. 

### 2.3. Post-Translational Modifications (PTMs)

Post-translational modifications (PTMs) induce chemical changes onto proteins that alter their function, interaction, and localization [57,58]. Given the dynamics and variety in PTMs [59,60], traditional methods of investigation make it difficult to study PTMs systematically. Fortunately, PTMs are strongly associated with enzymes and metabolic processes such as phosphorylation and methylation, which transiently regulate the fluxes through enzymatic reactions [61,62]. This provides an accessible framework for GEM integration to understand the role that PTMs play in cell networks in response to environmental changes. One area of focus is their role in bacterial metabolism, where PTMs serve as key metabolic regulators in response to different nutrition conditions [63].

*Regulated metabolic branch analysis* (RuMBA) introduced one way to study PTMs in bacterial cells [36]. This method analyzes how bacteria rewire their metabolic networks when exposed to different nutrients in their environment. RuMBA samples fluxes in the solution space to identify any areas in the metabolic network where major rerouting occurs. If the fluxes significantly change and result in the rerouting of a metabolite’s production, the associated reaction is identified as a “branch point” (Figure 3). Using RuMBA, the authors successfully unveiled enzymes that are critically regulated by PTMs, such as the fluxes tied to acetate switching from isocitrate dehydrogenase to isocitrate lyase due to phosphorylation. While RuMBA can identify enzymes regulated by PTMs, enzymes associated with regulations that occur over longer timescales require alternative methods. *Flux space shift* (FSS) analysis was developed to address this specific issue by focusing on differential enzyme regulation for optimal growth in certain nutrient conditions [64].

In contrast to RuMBA and FSS, *comparative analysis of regulators of metabolism* (CAROM) is a data-driven method that integrates omics data to decipher how both microbes and mammalian cells allocate different types of regulation, including PTMs [37]. CAROM applies FVA to obtain feasible flux ranges and evaluates regulation targets in the metabolic networks based on topological and flux properties. Specifically, CAROM revealed that enzymes with high topological connectivity are regulated by PTMs. CAROM also discovered that essential enzymes are regulated by acetylation, while those that catalyze reactions with high fluxes are regulated by phosphorylation. Both RuMBA and CAROM employ GEMs to examine flux solution spaces to effectively identify features important to PTMs. A key difference between these two methods is that while RuMBA uses flux information alone to predict PTM sites, CAROM uses data on enzyme, biochemical, and network topological properties. Hence, CAROM provides both a top–down systems perspective on regulation distribution between different PTMs and a bottom–up understanding of PTM regulation of specific enzymes.

The methods discussed above were specifically designed to study how PTMs influence metabolic activity by introducing new CBM constraints. In contrast, some other studies have investigated the role of PTM regulation in metabolism indirectly. For instance, *protein-specific information matrix* (PSIM) is a genome-scale model that focuses on the protein secretory system [65]. Given that PTM activity influences protein secretion and localization [66], the model accounts for signal peptides as well as N- and O-linked glycosylation. Indeed, PSIM was used to analyze the usage of cofactors and metabolic precursors for secretory PTMs occurring in both yeast and human cells. Another research extended the PSIM approach to model mammalian secretory systems [67]. This work curated three network models (human, mouse, and Chinese hamster ovary (CHO) cell) by integrating omics data and solving metabolic fluxes using FBA. Specifically, the authors modeled energy consumption for each secretory machinery and applied the curated models to optimize the production of monoclonal antibodies. Although these models mostly focused on secretory PTMs, they also included several other PTMs such as phosphorylation, which influences protein synthesis and cell growth. In the future, a model with both PTMs and transportation machinery may expand our understanding of metabolic regulation by PTMs.

### 2.4. Epigenetics

Epigenetics refer to biochemical changes that are transiently passed down to successive generations without modifying the DNA sequence. This enables eukaryotic cells to have identical genomes yet different phenotypes. While DNA cytosine methylation, non-coding RNAs, and histone modifications all play crucial roles in the mechanisms of epigenetics [68,69,70], histone modifications are the most sensitive to the metabolic state of the cell [71]. Histone proteins wrap around DNA and control gene transcription. Covalent modifications of histones, such as methylation and acetylation, can change gene expression without changing DNA sequences [72]. Since covalent modifications require small molecules and metabolites, histone modifications are highly influenced by metabolic states [73]. For example, metabolites including acetyl-coenzyme A (acetyl-CoA), S-adenosylmethionine (SAM), and nicotinamide adenine dinucleotide (NAD) are required to activate epigenetic enzymes [74]. Concentration levels for these three metabolites change via varying nutrient conditions and metabolic pathways, such as the TCA cycle, methionine metabolism, and glycolysis [75]. However, modeling covalent modifications on histones is a challenge because gene regulation and metabolic pathways are tightly connected [76].

GEMs potentially offer a framework for measuring how metabolism interacts with epigenetics [77,78]. However, a major limitation of using GEMs is the fact that key reactions involved in epigenetic regulation, such as those transporting or synthesizing metabolites such as Acetyl-CoA in the nucleus, are often missing. To address this problem, the *epigenome-scale metabolic network model* (EGEM) was developed [39]. This model incorporates acetylation reactions into the human GEM and emphasizes epigenetic effects by optimizing both biomass and acetylation production (Figure 3). EGEM accurately predicted bulk acetylation levels and the impact of disrupting histone acetylation on biomass using transcriptomics data from the Cancer Cell Line Encyclopedia (CCLE) database [79].

While EGEM mechanistically accounts for epigenetic regulation, recent studies have tried to model the resulting transcriptional impact of epigenetic regulation on metabolism. The *fast reconstruction of compact context-specific metabolic networks for the integration of transcriptomics data* (FASTCORMICS) method incorporates microarray data to existing GEMs [80]. Although FASTCORMICS is not designed specifically for epigenetics and does not simulate epigenetic-specific reactions as EGEM does, the reconstructed networks have successfully identified epigenetic features specific to each cell stage during the transition from human monocytes to macrophages. Similarly, Salehzadeh-Yazdi et al. incorporated transcriptomics data from histone tail mutant yeast strains to study acetylation in yeast [81].

As mentioned above, histone PTMs and metabolic states are naturally intertwined because of the small molecules required for covalent modifications. However, the process of epigenetic modification occurs in a highly dynamic fashion. Therefore, the changes in metabolite concentrations over time are important data that can provide better understanding of this interplay. The *dynamic flux activity* (DFA) approach constrains GEMs with time-course metabolomics [38,82]. Chandrasekaran et al. used a DFA model of stem cells and discovered that an increase in SAM synthesis, which supports histone methylation, occurs when mouse embryonic stem cells differentiate from naïve to primed state [38]. Although all the methods discussed above collectively helped us understand how metabolism influences epigenetic regulation and vice versa, they all specifically focused on acetylation and methylation. To further improve our understanding of epigenetic–metabolic interactions, future models should account for other epigenetic changes such as phosphorylation and the dynamic feedback regulation between these processes.

### 2.5. Protein–Protein Interactions and Protein Stability (PPIs/PS)

Enzyme activity is also influenced by the structural stability of the protein and its interactions with other proteins. Protein–protein interactions (PPIs), which entail physicochemical contacts between proteins, govern a wide range of cellular processes including but not limited to signal transduction [83], molecular transport [84], and cell metabolism [85]. Over the past two decades, millions of PPIs have been elucidated via both experimental and computational methods with varying degrees of confidence [86]. At the same time, these interactions have been annotated in multiple public databases that altogether provide PPI data for representative organisms from the animal, plant, and bacterial kingdoms [87]. Beyond their characterization, there has also been increasing interest in the therapeutic potential of targeting PPIs to treat cancer [88] and other diseases [89].

In the context of metabolism, PPIs play a major role in regulating reduction–oxidation (i.e., redox) reactions [90] and inducing enzymatic activity [91,92]. However, thus far, only one group has directly integrated PPIs into GEMs. Specifically, Lee et al. integrated TRNs and PPIs into GEMs for three human cell types: hepatocytes, myocytes, and adipocytes [42]. Using these models, the authors simulated changes in liver metabolism for obese individuals undergoing surgery and interestingly found that dysregulation in mannose metabolism correlated with insulin resistance. This same integrated modeling approach was leveraged in a successive study to elucidate metabolic dysregulation in other liver diseases [93].

Other than direct PPI integration into GEMs, several groups have integrated protein stability data to more accurately model metabolism using GEMs. Protein structural integrity is critical for proper functionality. One of the earliest studies by Chang et al. investigated *E. coli* growth and metabolism in extreme temperatures by integrating protein structure information [40]. This information included data on the amino acid sequence, native wild-type structure, functional annotation, and structural changes that occur upon protein–substrate binding. Using an extended *E. coli* GEM with protein structural information, Chang et al. not only simulated growth rate patterns resembling what was observed experimentally, but they also revealed metabolic mechanisms that may confer thermotolerance for heat-adapted *E. coli* strains. These findings were supported by follow-up experiments that showed increased growth for *E. coli* cultured in conditions supplemented with metabolites normally impacted by heat stress. 

A successive study by the same group formally introduced the computational process of integrating protein structure information into GEMs [41]. The approach, coined as “GEM-PRO” (Figure 3), was applied for *E. coli* and *Thermotoga maritima* to investigate how growth is limited by protein instability due to heat stress. The authors further showcased how GEM-PRO models can be used for applications at the intersection between systems and structural biology. However, GEM-PRO has not yet been widely applied since its introduction as it requires an abundance of protein structure information, which is not readily available nor attainable for most organisms. Although structural information may be derived from homology modeling, this method can often compromise the overall quality of the model.

### 2.6. Allostery

Allosteric regulation describes the modulation of enzyme activity via effectors, which are ligands or metabolites that bind to an enzyme at a location other than the active site. Upon binding to an enzyme, effectors can either enhance (i.e., activate) or attenuate (i.e., inhibit) activity through the associated reaction. This form of regulation is often depicted as “feedback” or “feedforward loops’’ depending on whether the effector metabolite is produced downstream or upstream of the reaction catalyzed by the regulated enzyme, respectively. Allosteric regulation enables biochemical adaptation to rapid changes in the cellular environment, particularly metabolite concentrations. Similar to PPIs/PS, allosteric regulation plays a key role in cell signaling, biomolecular transport, and metabolism [94,95].

Although allosteric regulation plays a major role in regulating metabolism, this regulatory aspect has not been widely considered in genome-scale metabolic modeling studies. In fact, Machado et al. was the first group to integrate genome-scale allosteric regulation with metabolic models [43]. Specifically, the authors developed a new constraint-based modeling method, called *allosteric regulation FBA* (arFBA), which was designed to incorporate allosteric regulation when solving for flux predictions. This method introduces a regulation (R) matrix (analogous to the S matrix) that characterizes effector–reaction relationships. A value of 1 indicates reaction activation via effector binding, while a value of −1 indicates reaction inhibition (Figure 3). In a tangential study, Hackett et al. directly used proteomic, metabolomic, and fluxomic data to model yeast metabolism and regulation across diverse conditions [44]. Their method, coined *systematic identification of meaningful metabolic enzyme regulation* (SIMMER), contextualizes experimental data via Michaelis–Menten rate law equations. The authors also used an extended *S. cerevisiae* GEM to infer fluxomics-constrained genome-scale metabolic fluxes via FVA. Both studies revealed distinct ways that metabolism is regulated depending on the environmental conditions. However, they both investigated allosteric regulation of metabolism at a small scale (≈100–300 reactions).

### 2.7. Signaling

The signaling pathways enable cells to quickly respond to fast-changing environmental and extracellular signals for a wide range of behaviors, from chemotaxis to developmental processes. Generally, a cascade of proteins participates in the process. For example, the Wnt signaling pathway plays an important role in embryonic development by controlling body axis formation [96]. This pathway is initiated when the Wnt protein binds to the membrane receptor Frizzled [97]. Then, this protein cluster phosphorylates downstream proteins to initialize a cascade of reactions regulating gene expression [96,97,98,99]. While a signaling network model can recapitulate the dynamics of a signaling pathway [100], this is not effective to systematically investigate how signaling networks interact with other pathways and lead to different phenotypic properties [101]. In this regard, GEMs are ideal candidates for modeling phenotypic changes and metabolic shifts driven by signaling. However, steady-state assumptions make it difficult to capture the dynamic nature of signal transduction, which is dependent on immediate metabolite concentrations (i.e., non-zero value allocation to the *b* vector in CBM). More importantly, signaling systems such as phosphorylation of the G-protein coupled receptor can respond to environmental perturbations quickly, but the reset of the system via receptor internalization can be slow [102]. In other words, reaction timescales are variable and consequently challenging to simulate [103,104].

To address this issue of timescales, metabolic modeling with ODEs enables the computation of metabolite concentrations as a function of time. However, this approach is limited by high computational costs, difficulty in finding unknown kinetic parameters, and the inability to connect to phenotypic properties. As with TRN mechanisms, dynamic kinetic modeling with ODEs can capture signaling mechanisms in ways that traditional modeling methods fail to achieve. To leverage the advantages of both GEMs and ODEs, two integrated models, iFBA [27] (discussed in the TRN section) and idFBA [45], were developed. Both methods applied ODEs to update the right-hand side of the LP problem (i.e., S×v=0) at each time step. Furthermore, idFBA employs the use of an incidence matrix to specify whether reactions are activated at a given discrete time step. The incidence matrix records binary values where 1 means the reaction is active at a given moment and 0 means it is inactive (Figure 3). Both iFBA and idFBA can model the dynamics of intracellular metabolites and reactions in the short timescale that is critical for signaling, but idFBA can better capture the changes from both fast and slow reactions. Nonetheless, both methods are limited by the size of their ODEs; that is, the complexity increases when incorporating more signaling networks. Moreover, constructing the incidence matrix for idFBA could be challenging if the timescales between reactions are significantly different or unclear, so a data-driven approach to define the matrix could improve the use of idFBA. Altogether, by integrating ODEs into GEMs, iFBA and idFBA partially overcome the limitations of the steady-state assumption, thereby shedding light on timescales relevant to signaling.

## 3. Areas for Improvement

While the methods discussed above represent key advances in the study of metabolic regulation using GEMs, there remain critical areas for improvement in modeling cellular regulation across the six highlighted mechanisms. One common area pertains to the generation and integration of high confidence experimental data. Specifically, the expansion of knowledge in environment- and condition-specific transcriptional regulation will be vital to broaden the scope of GEMs [105]. For modeling epigenetic effects, the integration of epigenetic modifications beyond acetylation, such as phosphorylation, is needed. For certain mechanisms, namely PPIs/PS and allostery, active databases containing regulatory annotations exist [87,106] but have yet to be widely integrated with GEMs. Of note, these databases may still lack the information required for successful GEM integration; however, such information may be supplemented by artificial intelligence (AI) methods predictive of molecular structure and regulatory interactions [37,107]. 

Integrated GEM model development involves an iterative process of incorporating regulatory information and conducting regular quality checks until a high-quality model is achieved. Once a draft model is reconstructed, the curation of these models poses additional challenges [19]. Errors may arise due to false positive interactions in the regulatory network or the metabolic network. *Gene Expression and Metabolism Integrated for Network Inference* (GEMINI) is the first approach for addressing this challenge of curating GEMs integrated with regulation [108]. GEMINI was able to prioritize false positive regulatory interactions in the PROM model of *S. cerevisiae* and greatly improve the model’s accuracy. Nevertheless, unlike the curation of traditional metabolic network GEMs, for which a comprehensive suite of tools exists, the curation of integrated GEMs has not been automated yet, and GEMINI has been applied only to TRNs.

Modeling regulatory mechanisms characterized by dynamic enzymatic activity, such as allostery and PTMs, will further require the inclusion of kinetic parameters that relate enzyme concentrations to metabolic flux changes. GECKO, or *GEM with enzymatic constraints using kinetic and omics*, is one method that could be leveraged to integrate enzyme kinetic information into GEMs [109]. This method integrated proteomics data and kinetic parameters into the *S. cerevisiae* GEM, yielding an updated model with kinetic constraints that generated more accurate simulations compared to previous models. Although GECKO demonstrated the value of incorporating kinetic constraints into GEMs, these parameters are not widely available for most enzymes. However, this limitation could be remedied by methods such as DeepEC (*Deep learning of Enzyme Commission numbers*) [110], which applies deep learning to determine enzyme kinetic properties directly from sequence information.

Another major area for improvement involves the ability to computationally model metabolism at the spatiotemporal level. This is because regulatory events often occur at different timescales (milliseconds vs. hours), and the degree of regulation can be distributed unevenly depending on the cellular objective [111]. For example, different rates of oxygen consumption within a cell colony lead to the spatially uneven distribution of regulatory events and reaction activity [112]. Spatiotemporal factors are also important to consider to holistically simulate metabolism, as multiple regulatory events tend to occur in tandem. Dynamic metabolic modeling could be achieved using methods such as *computation of microbial ecosystems in time and space* (COMETS) [113] and *3-dimensional dynamic flux balance analysis* (3DdFBA) [114], which simulate flux changes via partial differential equations (PDEs) and *dynamic FBA* (dFBA) [115]. It is important to note that the use of PDEs and dynamic constraints dramatically increases model complexity and computation time. However, the rapid advancement of computational capability and power may help to realize the spatiotemporal modeling of genome-scale metabolism.

## 4. Conclusions

Over the past three decades, GEMs have become integral for modeling metabolism at a systems-level perspective. GEMs have enabled the study of metabolic rewiring in response to environmental factors and paved the way for metabolic engineering in agricultural, industrial, and medical applications [15,116,117]. At the same time, advances in biochemical research have led to increased knowledge of the regulatory intricacies involved in a multitude of cellular processes.

Within this review, we covered six regulatory mechanisms relevant to metabolism: TRNs, PTMs, epigenetics, PPIs/PS, allostery, and signaling. We discussed how various methods have integrated these regulatory mechanisms into GEMs, thereby improving the accuracy in model simulations. We also highlighted areas for improving the integration of regulatory information into GEMs. These include the continued incorporation of highly curated experimental data on regulatory mechanisms, iterative curation of integrated GEMs, addition of dynamic constraints to capture spatiotemporal metabolic changes, and supplementation with AI methods that can provide missing information on protein structure [107], biomolecular interactions [37], and enzyme kinetic parameters [110].

Looking to the future of metabolic modeling, the successful construction of next-generation GEMs and ultimately whole cell models for complex organisms [118] relies on addressing three key areas. First, GEMs need to be expanded via the continued integration of known regulatory mechanisms. The methods discussed in this review have initiated the inclusion of regulatory information into GEMs but lack the continued integration of new information as it is made publicly available. Continuously updating GEMs to encompass the entirety of regulatory mechanisms at the genome scale will be a critical step in building next-generation GEMs. Second, next-generation GEMs will need to involve the simultaneous incorporation of multiple regulatory mechanisms. Models incorporating both TRNs and PPIs/PS have already shown tremendous value in personalized medicine via the construction of cell-specific networks used to design therapies for obesity [42]. Following this example, future GEMs will require the incorporation of multiple regulatory mechanisms to emulate regulatory mechanisms operating together to drive context-specific metabolism. Third, next-generation GEMs will need to model the crosstalk between different forms of regulation. Regulatory mechanisms do not operate independently but often influence each other in diverse ways. For instance, signaling pathways can activate TFs [119], allostery can modulate transcriptional regulators [120], and PTMs can affect protein stability and TF activity [121]. We ultimately believe that next-generation GEMs incorporating multiple regulatory mechanisms hold the potential for elucidating complex cellular interactions that drive human health and disease [122,123,124].

## Figures and Tables

**Figure 1 metabolites-11-00606-f001:**
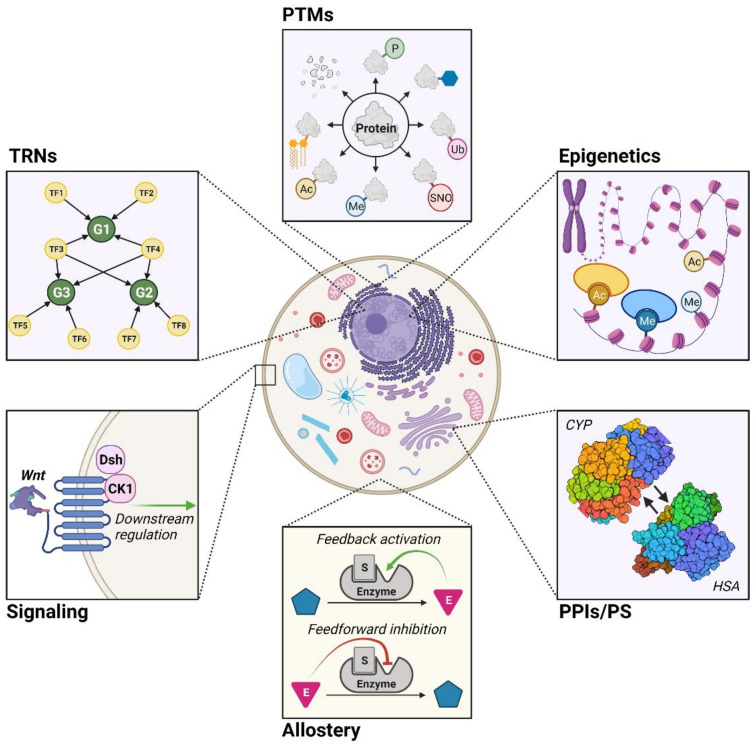
Six regulatory mechanisms that influence metabolism (bolded). **TRNs**: transcriptional regulatory networks, which describe how gene transcription is regulated (depicted: general TRN); **PTMs**: post-translational modifications, where proteins are enzymatically modified following their translation (depicted: phosphorylation (P), glycosylation, ubiquitination (Ub), S-nitrosylation (SNO), methylation (Me), N-acetylation (Ac), lipidation, proteolysis); **epigenetics**, which involve changes in gene expression without alterations the DNA itself (depicted: histone acetylation (Ac) and histone methylation (Me)); **PPIs/PS:** protein–protein interactions and protein stability, where functionality depends on direct protein–protein contact and their structural integrity (depicted: interactions between cytochrome P450 monooxygenase (CYP) and human serum albumin (HSA)); **allostery**, or the regulation of protein activity from non-active site ligand binding (depicted: general allosteric regulatory events); and **signaling**, which entails how signaling pathways govern the activity of a cell (depicted: Wnt signaling network).

**Figure 2 metabolites-11-00606-f002:**
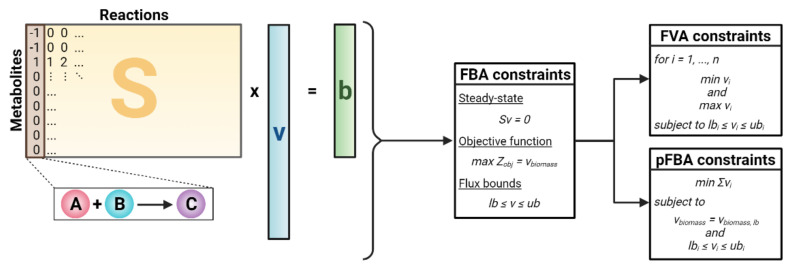
Mathematical framework of three constraint-based modeling (CBM) methods: flux balance analysis (FBA), flux variability analysis (FVA), and parsimonious FBA (pFBA). *S* = stoichiometric matrix, *v* = vector of reaction fluxes, *b* = vector of changes in metabolite concentration, *Z_obj_* = objective function, *v_biomass_* = biomass reaction flux, *lb* = lower flux bounds, *ub* = upper flux bounds.

**Figure 3 metabolites-11-00606-f003:**
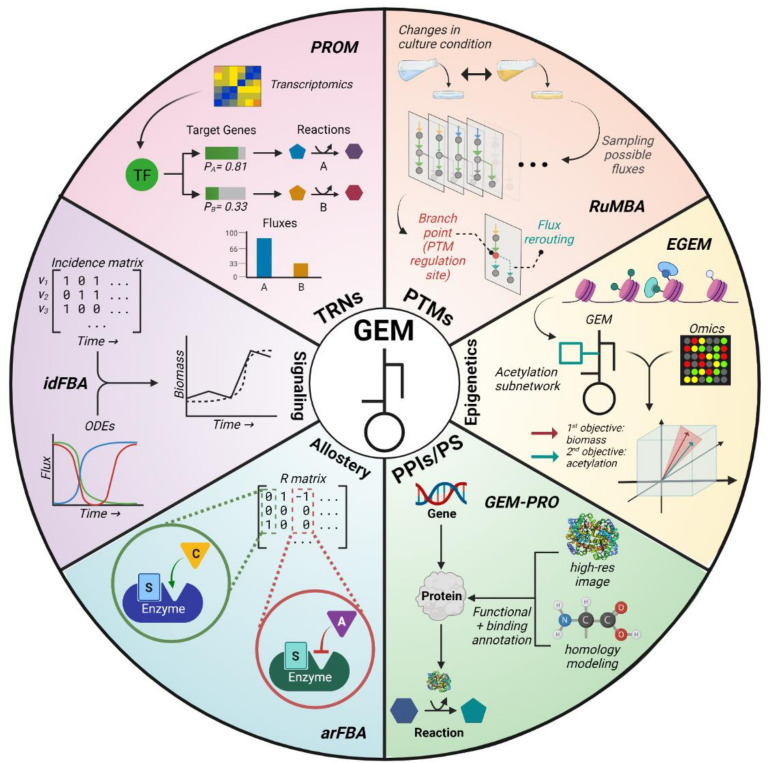
Representative algorithms (bolded) integrating regulatory mechanisms into genome-scale metabolic models (GEMs). **PROM**: *probabilistic regulation of metabolism*, uses transcriptomics and TF–gene networks to continuously restrict gene expression levels and then reaction fluxes [28]; **RuMBA**: *regulated metabolic branch analysis*, analyzes how fluxes change under different culture conditions to identify PTM regulatory sites [36]; **EGEM**: *epigenome-scale metabolic network model*, added a histone acetylation subnetwork to the human GEM and modified the objective function to maximize acetylation as well as biomass [39]; **GEM-PRO**: *genome-scale models with protein structure*, adds protein structural information into GEMs to capture how protein stability influences metabolic activity [41]; **arFBA**: *allosteric regulation flux balance analysis*, introduces a regulation (R) matrix that models the allosteric regulation of reactions in GEMs [43]; **idFBA**: *integrated dynamic flux balance analysis*, uses ODEs and an incidence matrix to model reaction fluxes dynamically [45].

**Table 1 metabolites-11-00606-t001:** 22 methods (bolded) integrating regulatory mechanisms into genome-scale metabolic models (GEMs). N/A: not applicable. Abbreviations: **rFBA**-regulatory flux balance analysis; **SR-FBA**-steady-state rFBA; **iFBA**-integrated FBA; **PROM**-probabilistic regulation of metabolism; **TIGER**-toolbox for integrating genome-scale metabolism, expression, and regulation; **IDREAM**-Integrated deduced and metabolism; **TRFBA**-transcriptional regulated FBA; **OptRAM**-optimization of regulatory and metabolic networks; **RuMBA**-regulated metabolic branch analysis; **CAROM**-comparative analysis of regulators of metabolism; **EGEM**-epigenome-scale metabolic network model; **GEM-PRO**-genome-scale models with protein structure; **arFBA**-allosteric regulation flux balance analysis; **SIMMER**-systematic identification of meaningful metabolic enzyme regulation; **idFBA**-integrated dynamic flux balance analysis

Method	Regulation	TRN Type	Year	Organism	Language	Summary	Ref.
**rFBA**	TRN	Boolean	2002	*E. coli*	MATLAB	Uses Boolean TRN to predict fluxes	[24]
**SR-FBA**	TRN	Boolean	2007	*E. coli*	MATLAB	Uses Boolean TRN to better characterize steady-state fluxes	[25]
**Lee et al.**	TRN	Discrete	2007	*E. coli*	LINGO + LabView	Integrates TRN with eight weight parameters to predict fluxes	[26]
**iFBA**	TRN/Signaling	Boolean	2008	*E. coli*	MATLAB	Uses Boolean TRN with kinetic parameters and ODEs to better predict fluxes	[27]
**PROM**	TRN	Continuous	2010	*E. coli,* *M. tuberculosis*	MATLAB	Uses transcriptomics and TF–target relationships to integrate a continuous TRN	[28]
**TIGER**	TRN	Boolean	2011	*S. cerevisiae*	MATLAB	Integrates TRN + GEM + transcriptomics	[29]
**FlexFlux**	TRN	Boolean/ Continuous	2015	*E. coli*	Java	Integrates TRN + GEMs in SBML format	[30]
**PROM 2.0**	TRN	Continuous	2015	*M. tuberculosis*	MATLAB	Uses transcriptomics and TF–target relationships to integrate an expanded continuous TRN	[31]
**CoRegFlux**	TRN	Continuous	2017	*S. cerevisiae*	R	Predicts fluxes with reverse-engineered TRN	[32]
**IDREAM**	TRN	Continuous	2017	*S. cerevisiae*	MATLAB	Predicts fluxes with continuous reverse-engineered TRN	[33]
**TRFBA**	TRN	Continuous	2017	*E. coli,* *S. cerevisiae*	MATLAB	Uses transcriptomics and TF–target relationships to more intuitively integrate a continuous TRN	[34]
**OptRAM**	TRN	Continuous	2019	*S. cerevisiae*	MATLAB	Strain design algorithm that uses IDREAM	[35]
**RuMBA**	PTMs	N/A	2018	*E. coli*	MATLAB	Identifies branch-point reactions regulated by PTMs via flux sampling	[36]
**CAROM**	PTMs	N/A	2019	*E. coli,* *S. cerevisiae*	MATLAB	Integrative analysis of multi-omics data to predict PTM regulation	[37]
**Chandrasekaran et al.**	Epigenetics	N/A	2017	Stem cell	MATLAB	Uses time-course metabolomics data to infer fluxes, such as those involved in methylation	[38]
**EGEM**	Epigenetics	N/A	2019	Cancer cell	MATLAB	Simulation of multi-objective model with an acetylation subnetwork	[39]
**Chang et al.**	PPIs/PS	N/A	2013	*E. coli*	MATLAB	Integrated protein binding and structure information into the *E. coli* GEM	[40]
**GEM-PRO**	PPIs/PS	N/A	2016	*E. coli,* *T. maritima*	Python	Describes general process of integrating protein information into GEMs	[41]
**Lee et al.**	PPIs/PS	N/A	2016	Liver cells	MATLAB	Integrated TRNs and PPIs to construct cell-specific networks to study liver metabolism	[42]
**arFBA**	Allostery	N/A	2015	*E. coli*	Python	Integrates allosteric interactions into GEMs	[43]
**SIMMER**	Allostery	N/A	2016	*S. cerevisiae*	R	Accounted for allosteric regulation but mostly relied on ODE modeling	[44]
**idFBA**	Signaling	N/A	2008	*S. cerevisiae*	MATLAB	Incorporates ODEs and an incidence matrix to model dynamics	[45]

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
