# Peer review of "Next-Generation Genome-Scale Metabolic Modeling through Integration of Regulatory Mechanisms"

_metabolites, 2021, doi:10.3390/metabo11090606_

Round 1
Reviewer 1 Report
The authors present here a very relevant and welcome review detailing the efforts devoted so far to integrate regulatory mechanisms into genome-scale models (GSMs). GSMs are powerful tools for documenting, understanding, and predicting metabolism. However, most of them fail to account for the regulatory mechanisms that are so critical to understand metabolism.
Here, the authors present a comprehensive review of the efforts to integrate this information into GSMs. I find the review well written, comprehensive, and of invaluable help to the metabolic modeler. Hence, I recommend its publication, but with some changes:
- Line 60: it would useful to cite some specific examples of GSMs failing. There is a dearth of reported negative results, which is hindering the whole field.
- Line 116: Ref 41 by Schuetz et al. does not only use biomass optimization, as implied by the way the text is written.
- Line 119 is cryptic. This reviewer cannot make sense of it. Please rewrite and make clearer. Perhaps split into two sentences?
- Equations 3.1 and 3.2 are not correct. Minimization and maximization are subject to an extra constraint due to the objective function value.
- Line 125: pFBA is not an alternative to FVA. Whereas FVA aims to find lower and upper bounds for the fluxes, pFBA aims to find the smallest fluxes compatible with the obtained objective function value (e.g. growth rate). They should not be conflated.
- Line 130: the results achieved using pFBA tend to emphasize a small number of the high-flux reactions by construction. This is not a coincidence, and should not be presented as such.
- Caption for Figure 3 could be more informative and add at least a small description of each method.
Reviewer 2 Report
Thank you for the opportunity to review this work. The manuscript under this review discusses integrative Genome-scale metabolic models (GEM) modeling methods and how these have been used to simulate metabolic regulation during normal and pathological conditions.
The study is interesting and original. However, there are some concerns that need to be addressed before the manuscript could be considered for publication in Metabolites.
Abstract
The presentation of the abstract is clear. Congrats.
Introduction
Well-presented and written, but they should increase the number of references that support the theoretical framework presented and improved several concerns.
Lines 31-32: I suggest the authors add a bibliographic citation to support this cellular metabolism.
Lines 34-36: Similarly to suggested in the previous sentence, support with bibliographic citations.
Line 65: Figure 1. I suggest the authors move the figure to this line and the six different types described in lines 66-76 be explained within the same figure as a) TRNs, b) PTMs, c) Epigenetics, d) PPIs/PS, e) Allostery and f) Signaling in the corresponding figure caption.
Line 82: The citations of the 22 signaling methods described in Table 1 are missing in the text.
Modeling of Metabolic Regulation
2.1. Simulating metabolic networks using CBM
Line 94: Consider attaching figure 2 just after this point the first time it is named in the text.
Line 109: Please indicate Zobj in parentheses the first time it is named in the manuscript.
Equation 1, indicate in the text its location and corresponding explanation.
Equations 2.1 and 2.2, likewise, indicate their corresponding explanation in the text of the manuscript.
Equation 4, indicate in the text its location and corresponding explanation.
2.2 Transcriptional Regulatory Networks (TRNs)
Line 139: I suggest the authors, the first time they name mRNA, explain its full meaning.
2.2.2. Continuous TRNs
Line 198: After this paragraph, I suggest that the authors please place figure 3, being the first time, it is presented and in subsequent paragraphs, only present the figure in the text as it is presented.
I suggest that in the text state the exact p-values shown in figure 2.
2.3 Post-Translational Modifications (PTMs)
Lines 227-229: After these two sentences presented by the authors, consider supporting with relevant bibliography.
Lines 252-260: This paragraph is exempt from citations that support these judgments offered by the authors. I suggest they include references that support these claims.
2.4. Epigenetics
Lines 280: I suggest the authors, the first time they name DNA, explain its full meaning.
2.7. Signaling
Line 396: What does Wnt signaling mean? I suggest that the authors explain the full meaning for a better understanding throughout the manuscript.
Areas for Improvement
I congratulate the authors for the clarity of the explanation in this section.
Conclusion
Lines 478-479: The authors indicate that they have presented six relevant regulatory mechanisms in metabolism. However, in the development of the manuscript there are 7 points, including 2.1. Simulating metabolic networks using CBM (lines 92-136). Could you explain why only 6 mechanisms are presented and 7 are developed in the manuscript?
I congratulate to authors for the manuscript present, and I hope that the suggestions presented will help to improve the manuscript submitted for publication in Metabolites, wishing you all the best in the future.
Reviewer 3 Report
This manuscript is a well-written review on the integration of regulatory mechanisms into GEMs. My main concerns are about the figures and some definitions that hopefully can improve the readability of the manuscript.
Figure 1:
- TRN should be a directed network.
- it would be nice if the authors depict actual signaling pathways, not just their functionalities (activation and inhibition)
- Elaborate on the epigenetics modifications definition, if the authors focused on the epigenetic modifications at the gene expression level, they should also mention Micro-RNAs, otherwise please replace them with histone modifications.
- For the allosteric modulation, I suggest using Negative and Positive feedback instead of feedback and feedforward.
Figure 2:
- S, should be a matrix of numbers (please draw an actual matrix with rows and columns and some numbers)
- What a reader should imply from the box below the S matrix?
- b, in CBM must be zero.
Figure 3:
- It is hard to follow. It could be more informative and accurate.
It would be nice if the authors also mention new platforms that have been developed for GEMs analysis, such as GECKO (https://www.embopress.org/doi/full/10.15252/msb.20167411)
It would be nice if the authors also mention some benchmarking investigations in this domain and summarize them. It is also helpful for the conclusion section.
In the conclusion section, the take-home message is missed. If a user wants to integrate regulatory mechanisms into GEMs, what are the author's suggestions?
Round 2
Reviewer 3 Report
The authors have made a great effort to address all the comments from all reviewers. I believe the manuscript has substantially improved and now meets the required quality for publication.
Therefore, my recommendation is to publish it in its present form.